# Will Anti-Epidemic Campus Signals Affect College Students’ Preparedness in the Post-COVID-19 Era?

**DOI:** 10.3390/ijerph18179276

**Published:** 2021-09-02

**Authors:** Teng Zhao, Yuchen Zhang, Chao Wu, Qiang Su

**Affiliations:** 1Zhejiang Academy of Higher Education, Hangzhou Dianzi University, Hangzhou 310018, China; zhaoteng@hdu.edu.cn; 2Chinese Academy of Science and Education Evaluation, Hangzhou Dianzi University, Hangzhou 310018, China; zyc@hdu.edu.cn; 3Propaganda Department, Hangzhou Dianzi University, Hangzhou 310018, China; we926@hdu.edu.cn

**Keywords:** COVID-19, campus signal, disaster preparedness, disaster awareness, structural regression model

## Abstract

The COVID-19 pandemic has been a tremendous global threat and challenge for human beings, and individuals need to be prepared for the next wave of the outbreak, especially in the educational setting. Limited research has focused on individual knowledge, awareness, and preparedness of COVID-19 in postsecondary institutions in the post-COVID-19 era so far. This study aimed to explore whether students’ perceived anti-epidemic campus signals had effects on their awareness of and preparedness for COVID-19. Leveraging the data collected from full-time college students in a province located in East China and building a structural regression model, we found that students’ perceived anti-epidemic campus signals were significantly associated with their awareness of and preparedness for COVID-19. With one perceived signal decrease, there were 0.099 unit and 0.051 unit decreases in students’ awareness and preparedness, respectively. In addition, we indeed found that female students had a higher awareness and better preparedness than their male peers. These findings provided important implications for postsecondary administrators and policymakers, as well as future research.

## 1. Introduction

The COVID-19 pandemic has been a tremendous global threat and challenge for human beings. According to the Coronavirus Resource Center at Johns Hopkins University, to date, more than 172 million cases and 3.7 million deaths have been reported across the world [1]. The strong infectiousness requires people to keep social distance, which has challenged the education system [2]. All levels of educational institutions, such as secondary schools and postsecondary institutions, were shut down temporarily and shifted to online courses during the peak of the pandemic. With the massive efforts endowed by governments and the people, COVID-19 in China has been effectively controlled, and the education system has also returned to normal operation. However, since the pandemic is still far from over, necessary preparations to prevent the infection and spread of the virus are needed.

Previous research has found that physical school environments could significantly affect student behavior [3,4]. For example, Johnson found that a positive school environment can increase perceived fairness and, thus, reduce school violence [5]. As such, college students’ perception of the anti-epidemic tension released by universities may influence their awareness of the pandemic. This is also partially consistent with Tkachuck et al.’s view that students’ perceived university preparedness was positively associated with disaster concern [6].

In China, as the majority of college students live on campus, they are likely to observe and experience campus changes, such as campus policy changes (e.g., social distance policy and health code policy) and physical environment changes (e.g., separators on the dining tables and banners in campus cafeterias). These are signals that deliver important campus messages to students to inform them that something may happen. Especially at this particular time, universities and colleges require students to stay on campus more, which itself could be regarded as a signal that students still need to be aware of the pandemic.

The reality is, as China quickly enters the post-COVID-19 era, though the overall anti-epidemic work is effective, there are still sporadic and recurrent outbreaks in some places, which periodically alert the whole nation. Campus anti-epidemic policies and/or measures are appropriately adjusted in terms of the actual national situation of the epidemic. When the pandemic is well controlled, in general, campuses have loose anti-epidemic actions; inversely, if the pandemic has signs of resurgence, campuses will immediately be put on alert and take anti-epidemic actions. In this study, we refer to these kinds of anti-epidemic actions as anti-epidemic campus signals that deliver messages to students to inform them of the severity of the pandemic. These campus signals are likely to influence students’ preparation awareness of COVID-19. For instance, when a campus has strict anti-epidemic policies, it may increase students’ preparation awareness. On the contrary, a campus with undemanding anti-epidemic policies may decrease it.

This study seeks to better understand whether anti-epidemic campus signals affect college students’ preparedness of COVID-19. This is crucial for research literature, as well as having practical implications. Past studies only theoretically suggest how educational institutions prevent COVID-19 [2] and suggest recommendations for medical students’ COVID-19 preparedness [7,8]. To our knowledge, there is no existing literature that analyzes the relationship between anti-epidemic campus signals and students’ COVID-19 preparedness. As a consequence, there is insufficient empirical evidence to inform postsecondary administrators and policy makers of whether anti-epidemic campus signals contribute to increasing students’ COVID-19 preparedness, even in a relatively safe situation. Leveraging the data collected from the survey of College Students’ Epidemic Preparedness in Post-COVID-19 Era (CSEPPCE), we examine the relationship between campus signals and college students’ preparedness with structural equation models.

## 2. Literature Review

### 2.1. Research on Knowledge, Awareness, and Preparedness in Disaster Management

Researchers have documented how preparedness is important for disaster management [9]. Experiences from past disasters affirm that pre-disaster preparedness can effectively reduce the loss brought by disasters and shorten milling processes. For example, the 1994 Northridge quake in Southern California that quickly reached a stable condition largely relied on residents’ being well-informed about earthquake preparedness [10]. In fact, better grasping the knowledge of disasters, such as what the disaster is and what the mechanism of the disaster is, and having a stronger awareness of disasters, such as the perception of risk, contribute to better preparedness.

The existing research has found that disaster knowledge and awareness of disasters are strongly associated with disaster preparedness [11,12,13,14]. Yu et al. applied a moderated mediation model using data collected from 1080 villagers in Shanxi province and found that the positive relationship between villagers’ disaster preparedness and communication with local officials was mediated by their disaster knowledge [11]. Using residents’ data from the hazard-threatened areas located in the Three Gorges Reservoir area and conducting regression models, Xu et al. found that residents with higher risk perception were more likely to adopt preparedness for a sudden landslide [15].

Specifically, in the circumstance of our study, COVID-19, we hypothesized the following:

**Hypothesis** **1.***Students who have better knowledge of COVID-19 will be better prepared*.

**Hypothesis** **2.***Students with higher awareness of COVID-19 will be better prepared*.

Comprehensively understanding disaster risks, one of the categories of disaster knowledge [16], ought to increase individuals’ awareness. The relationship between disaster knowledge and awareness has also been investigated in the prior literature. Interviewing 50 secondary students, Pinar suggested that disaster education that contributes to students’ knowledge of the disaster needs to be carried out by different stakeholders in order to raise their awareness [17]. Furthermore, some literature [18,19,20] has focused on disaster education programs and/or training, which can increase students’ knowledge of disasters, to examine whether disaster knowledge is associated with awareness. For instance, Ozkazanc and Yuksel found that students who received disaster training have a significantly higher awareness of disasters [20].

Specifically, in the circumstance of our study, COVID-19, we hypothesized the following:

**Hypothesis** **3.***Students who have a better knowledge of COVID-19 will have a higher awareness of COVID-19*.

In addition, disaster knowledge, awareness, and preparedness may also vary among different groups of the population. Eisenman et al. illustrated that Latinos were at a relatively lower level of disaster knowledge and preparedness [21]. Elliott and Pais found that Blacks were more likely to be evacuated after the storm, rather than before the storm, compared to similar Whites [22], indicating that Blacks may have a lower risk perception or awareness. According to these findings, it seems that socially underrepresented groups have a lower disaster knowledge, awareness, and preparedness. Kohn et al. reviewed 36 studies and found that different groups of the population had different disaster preparedness [12]. Cvetković et al. found that men perceived greater preparedness at both the individual and household level after the flooding [23].

Specifically, in the circumstance of our study, COVID-19, we hypothesized the following:

**Hypothesis** **4.***Male students have a higher awareness of COVID-19 and are more likely to be better prepared than their female peers*.

**Hypothesis** **5.***Han students, as the socially dominant group in China, have a higher awareness of COVID-19 and are more likely to be better prepared than their peers of other ethnicities*.

Guided by the above literature, our study examines whether the relationships between knowledge and preparedness, between awareness and preparedness, and between knowledge and awareness also apply to college students at this particular moment in the post-COVID-19 era. Additionally, we investigated whether there are variations between different gender groups and ethnicity groups. This study is extremely important because gatherings in postsecondary institutions may accelerate the spread of COVID-19, and better understanding the factors that influence students’ preparedness for COVID-19 is crucial. Considering COVID-19 is a type of disaster, we expected a similar pattern in these relationships compared to other types of disaster, such as landslides and earthquakes.

### 2.2. Research on Signals, Awareness, and Preparedness in Disaster Management

Studies exploring the effect of signals on individuals’ awareness and preparedness are relatively scarce. The majority of this research has focused on warning signals and has aimed to analyze the influence of such signals on individuals’ awareness and preparedness [24,25,26]. These warning signals delivered to the public through media contribute to people’s perception of disaster risks and preparedness [26]. Currently, generally, if a disaster can be detected, warning signals will be provided by authorities before the disaster. However, unlike other disasters, COVID-19 outbreaks cannot be precisely predicted. A better approach is to increase risk perception and regular anti-epidemic preparedness [27]. In an extreme scenario, if authorities enforce the public to adopt anti-epidemic measures, such as wearing masks, the public may increase in COVID-19 awareness and make adequate preparations.

From a business management perspective, Bazerman and Hoffman elaborated that individuals can be affected by their perceptions of the environment of the organizations to which they belong [28]. Similarly, in an education setting, college students in postsecondary education may perceive the anti-epidemic campus context, and their awareness and preparedness of COVID-19 may be influenced. Little was known as to whether anti-epidemic measures taken by campuses would influence students’ awareness and preparedness for the next potential wave of the outbreak. The anti-epidemic measures required by campuses have changed in terms of COVID-19, and this provides a unique opportunity to investigate whether the anti-epidemic campus signals influence students’ awareness and preparedness of COVID-19. Thus, we hypothesized the following:

**Hypothesis** **6.***A perceived reduction in anti-epidemic campus signals will decrease their awareness of COVID-19*.

**Hypothesis** **7.***A perceived reduction in anti-epidemic campus signals will weaken their preparedness*.

Filling the gaps in the literature, the purpose of our study is to evaluate the impact of anti-epidemic campus signals on students’ preparedness of COVID-19, and to examine the relationships among knowledge, awareness, and preparedness in the COVID-19 pandemic in an educational setting. In other words, we seek to test the aforementioned hypotheses.

## 3. Methods

### 3.1. Data Sources and Participants

Guided by Ikhlaq et al. [29] and Ahmed et al. [30], we carefully designed the survey, entitled *College Students’ Epidemic Preparedness in Post-COVID-19 Era*, which was randomly distributed to full-time college students in a province located on the east coast of China. The questionnaires mainly focused on students’ knowledge, awareness, and preparedness of COVID-19, as well as the anti-epidemic campus signals they perceived. Several items were designed to measure students’ knowledge, awareness, and preparedness to more precisely represent these complex and multifaceted variables (see details in Table 1).

The survey was distributed to 1600 full-time college students from 13 postsecondary institutions on May 2021 via WJX.CN, a platform that enables individuals to design surveys and then share survey links to intended participants. We recruited faculties from these postsecondary institutions to help us to share the survey link. The final response rate was 91.88%, yielding a sample of 1470. There were only a few responses to certain survey items missing, so we applied a listwise deletion technique to deal with such missingness, yielding a final analytic sample of 1464. Among them, 690 (47.13%) students were female, and 774 (52.87%) were male. In terms of ethnicity, 1372 (93.72%) were Han, and 92 (6.28%) were of other ethnicities.

### 3.2. Measures

#### 3.2.1. Endogenous Variables

In our model, one endogenous variable, students’ preparedness for COVID-19 in the current period, was measured by three survey items; that is, “will you wear a surgical mask when going outside?”, “will you use hand sanitizer?”, and “will you keep social distance when you are in public?”. They were all constructed on a five-point Likert scale with 1 = “Never” and 5 = “Always”. The mean of each item was 2.21, 2.24, and 2.23, respectively, which ranged between “Sometimes” and “Neutral” (see Table 2).

Another was students’ awareness of COVID-19, which was measured by seven items on a five-point Likert scale in the survey. However, after the initial check of correlations among these items, four items with low correlations were dropped from the analysis. The remaining three items were “how often will you talk about COVID-19 with your classmates”, “how often will you talk about COVID-19 with your friends”, and “how often will you talk about COVID-19 with your family”, where means = 2.37, 2.26, and 2.29, respectively.

#### 3.2.2. Exogenous Variables

The perceived change of anti-epidemic campus signals was measured by students’ perceived anti-epidemic campus signals in the peak of the outbreak minus students’ perceived anti-epidemic campus signals in the current stage. When designing the survey, we carefully checked with recruited faculties on the anti-epidemic campus signals that their own institutions had and selected representative signals that closely related to students’ daily life to ensure that surveyed students would easily recognize these signals. These included separators on the dining table, anti-epidemic banners, hand sanitizer in public areas, strict leaving school management, social distancing during classes, the integration of online and face-to-face classes, social distancing in public areas, and school health codes. Different campuses have different anti-epidemic measures, so students from different campuses may experience different anti-epidemic signals. Even on the same campus, the anti-epidemic tension perceived by students may be different. Thus, it is reasonable to use students’ perceived change of anti-epidemic campus signals to represent the perceived anti-epidemic tension on campus. Table 2 shows that the mean of students’ perceived change of anti-epidemic campus signals was 2.76, indicating that students did feel that campus anti-epidemic measures were suspended in the post-COVID-19 area when compared to the peak of the outbreak.

Students’ knowledge of COVID-19, as another exogenous variable, comprised four survey items, also on a five-point Likert scale. We asked: “whether you know about the symptoms of COVID-19”, “whether you know how COVID-19 is spread”, “whether you know the anti-epidemic measures of COVID-19”, and “whether you know the differences between COVID-19 and other pandemics such as SARS”, with 1 = “Not at all familiar” and 5 = “Extremely familiar”. The mean of each item was 2.93, 3.07, 3.16, and 2.66, respectively.

### 3.3. Analytic Plan

First, descriptive statistics were provided as in Table 2 to display the basic information about variables, using Stata 16. Second, correlations between each variable were analyzed to help us select appropriate survey items to measure our latent variables (i.e., knowledge, awareness, and preparedness).

Third, we adopted a two-step approach, suggested by Anderson and Gerbing [31], to assess the fitness of our full structural regression model using Mplus 8 [32]. In the first step, the measurement model was evaluated to assess its adequacy. Applying a diagonally weighted least squares estimator with mean and variance adjusted (WLSMV) based on the polychoric correlation matrix [33], we conducted step-wise measurement model comparisons until the final measurement model was adequate, judging by model fit indices such as the root mean square error of approximation (RMSEA), the standardized root means square residual (SRMR), the comparative fit index (CFI), and the Tucker–Lewis fit index (TLI). In addition to these indices, we also performed a chi-square (*χ*^2^) different test to compare which model was a better fit and then proceeded to the second step. In the second step, we evaluated the adequacy of structural components by comparing the fitness of the structural model to the fitness of the final measurement model until it was adequate. The major relationships that were tested were: (1) the perceived change in campus signals and awareness, (2) the perceived change in campus signals and preparedness, (3) knowledge and preparedness, (4) knowledge and awareness, and (5) awareness and preparedness.

Fourth, in order to understand whether students’ demographic characteristics, such as gender and ethnicity, would also influence students’ preparedness of COVID-19, we further set gender and ethnicity as control variables in the final structural regression model.

## 4. Results

### 4.1. Correlation Results

Table 3 showed the correlations among survey items. Among them, Items 2–5 demonstrated high correlations with a range from 0.69 to 0.87. The correlations among Items 6–8 were from 0.81 to 0.87, and Items 9–10 were from 0.56 to 0.61. These relatively high polychoric correlations provided evidence for us to use these survey items to measure our latent factors: knowledge, awareness, and preparedness, respectively.

### 4.2. Two-Step Approach Results

#### 4.2.1. Measurement Model Results

Following Anderson and Gerbing’s two-step approach [31], we then tested the measurement model associated with our full structural model, denoted as the initial model in Table 4. According to the cutoff values of the model fit indices recommended by Hu and Bentler [34], the initial model demonstrated a reasonable fit with *χ*^2^ (42) = 444.86, CFI = 0.990, TLI = 0.988, RMSEA = 0.081, and SRMR = 0.051. The model modification indices provided by Mplus software suggested correlations between errors of certain survey items. Upon the consideration of the real meaning of survey items, knowing how COVID-19 is spread helps to know anti-epidemic measures, we applied a step-wise measurement model comparison procedure by adding the correlated error between Items 3 and 4 into our first adjusted measurement model, yielding a better model fit with *χ*^2^ (41) = 403.14, CFI = 0.991, TLI = 0.989, RMSEA = 0.078, and SRMR = 0.050. Considering that students who are well informed about COVID-19 may know the differences between COVID-19 and other pandemics, we then added the correlated error between Items 2 and 5 into the first adjusted measurement model, yielding an even better model fit with *χ*^2^ (40) = 374.37, CFI = 0.992, TLI = 0.989, RMSEA = 0.076, and SRMR = 0.048.

To further affirm which model was statistically better, we conducted a chi-square difference test between the initial model and the adjusted models. We found that in the third model with the correlated errors of Items 3 and 4 and Items 2 and 5, compared to the initial model, the chi-square difference was Δ*χ*^2^ = 70.49, *p* < 0.01, indicating that the third model outperforms the initial model.

#### 4.2.2. Structural Model Results

After we were satisfied with our adjusted measurement model, we evaluated the structural part of the full structural regression model. The full structural model yielded a good model fit with *χ*^2^ (38) = 119.67, CFI = 0.998, TLI = 0.997, RMSEA = 0.038, and SRMR = 0.022. To assess the fit of the structural part of the model, we also conducted a chi-square difference test and obtained Δ*χ*^2^ = 254.7, *p* < 0.01 (See Table 4), indicating that the structural part of the model was adequate.

We acquired our final full structural regression model, as presented in Figure 1. Figure 1 shows that the standardized factor loadings for students’ knowledge of COVID-19 ranged from 0.819 to 1.013. Jöreskog pointed out that standardized factor loadings can exceed 1.00 and do not necessarily imply a wrong result [35]. The ranges of standardized factor loadings were from 0.880 to 0.947 and from 0.722 to 0.792, respectively, for students’ awareness of COVID-19 and students’ preparedness for COVID-19. These relatively high factor loadings suggested that three latent factors were well explained by the respective survey items. For example, for the item “know how COVID-19 is spread”, the corresponding r square value was 0.671, meaning that 67.1% of the variance in “know how COVID-19 is spread” was explained by students’ knowledge of COVID-19.

Looking at the paths in the structural regression model, we found that one unit increasing in students’ knowledge of COVID-19 was associated with a 0.210 unit increase in students’ preparedness for COVID-19. Thus, Hypothesis 1 was accepted. We also found that students’ awareness of COVID-19 was positively and significantly associated with their preparedness (β = 0.310, *p* < 0.001), which supported Hypothesis 2. Furthermore, the relationship between students’ knowledge and preparedness was also positively statistically significant (β = 0.232, *p* < 0.001); therefore, Hypothesis 3 was accepted.

Focusing on our primary interest of variables, students’ perceived anti-epidemic campus signals, we found that when students perceived that an anti-epidemic campus signal was decreasing, there was a 0.099 decrease in their awareness of COVID-19, which proves Hypothesis 6. Not surprisingly, a signal perceived as decreasing by students was associated with a 0.051 decrease in their preparedness. Thus, Hypothesis 7 was supported.

### 4.3. Structural Regression Model with Controls

As shown in Table 5, when controlling gender and ethnicity in the structural regression model, the model fit indices indicated a good fit, with *χ*^2^ (51) = 205.33, CFI = 0.997, TLI = 0.995, RMSEA = 0.045, and SRMR = 0.021. Compared with the model without gender and ethnicity controls (Model 1), the path coefficients of the model with controls (Model 2) remained statistically significant and in the same directions. Focusing on gender, male students had a lower awareness (β = −0.163, *p* < 0.01) and preparedness (β = −0.113, *p* < 0.01) of COVID-19 than their female peers. No evidence was found that students’ ethnicities would contribute to their awareness and preparedness. Therefore, both Hypotheses 4 and 5 were rejected.

## 5. Discussions

Building upon the previous literature, this study utilized the survey data to investigate how anti-epidemic campus signals affect students’ preparedness for the COVID-19 pandemic. Signals have been found to make individuals aware of risks before and/or after disasters [26,36]. In an educational setting, we indeed found that these signals could increase students’ awareness and contribute to a better preparedness for COVID-19. In addition to this, we also found that students who had more knowledge of COVID-19 were more likely to have a higher awareness and declare better preparedness. In addition, we found that students with a higher awareness were more likely to have better preparedness. Notably, gender was observed to have an influence on students’ awareness and preparedness, while ethnicity was not.

Cahapay believed that preparedness for COVID-19 is a priority of education in the post-COVID-19 era [37]. Exploring the factors that influence students’ preparedness of COVID-19 is crucial. Our findings showed that students’ knowledge and awareness all statistically significantly predict their preparedness. These findings are consistent with Nindrea et al. [38], showing that breast cancer patients’ COVID-19 knowledge and awareness were significantly associated with their preparedness. With respect to anti-epidemic campus signals, to our knowledge, no study has focused on this. However, Peng et al. demonstrated that the national anti-epidemic measures effectively help to reduce the reported number of confirmed cases [39], which helps to show that individuals are prepared for COVID-19, partially due to the perceived severities of the pandemic, which is a result of the nation’s response to the pandemic. As such, if postsecondary institutions implement strict anti-epidemic measures on campuses, students may have a stronger awareness of COVID-19 and make better preparations. This is supported by our structural regression model.

With respect to the demographic characteristics’ influence, past studies have found that gender and race/ethnicity have impacts on disaster management [40]. For example, Teo found that there were significant variations in disaster preparedness among different ethnicity groups [41]. Cvetković et al. found that men were more prepared than women in flooding events [23]. Although these results are not consistent with our findings, it may be due to the different types of disasters. It is possible that men feel that they are physically stronger and, thus, are more prepared for flooding, while they are less prepared for the pandemic because they may think that if they are infected, they will not get sick. This may be partly why we found that women were more prepared than men in the COVID-19 pandemic. Our finding that there is no impact of ethnicity on COVID-19 awareness and preparedness may be due to the small sample size and that the majority of students were Han.

The findings of this study contribute to the literature and have practical implications. This study is probably the first to look at the effects of anti-epidemic campus signals on students’ COVID-19 awareness and preparedness. This study shows how postsecondary institutions’ actions can influence students’ awareness and preparedness in the post-COVID-19 era. In addition, it provides further empirical evidence that COVID-19 knowledge can contribute to both awareness and preparedness, and awareness can influence preparedness. Based on these results, we recommend that postsecondary administrators focus on building a tense anti-epidemic atmosphere by establishing more anti-epidemic measures. Students living on campuses without strict anti-epidemic measures may fail to prepare for COVID-19. Once a student is infected, the virus is likely to spread quickly due to the gatherings that occur in campus settings. Thus, it is crucial to increase students’ risk perceptions of the pandemic and to be better prepared for future pandemics. Better preparation helps to avoid the kind of losses that occurred at the beginning of 2020. Notably, the findings could also inform policymakers of how to make policy decisions at both the institutional and individual level to better prepare for future pandemics.

The structured questionnaires were carefully designed, and the effect of anti-epidemic campus signals on students’ COVID-19 preparedness was comprehensively examined, but this study still has the following limitations. First, there are still unobserved variables, such as the whole nation’s anti-epidemic measures, which are also likely to influence the public’s preparedness but cannot be directly measured. These unobserved variables are likely to influence the model results but were inevitably tested. Second, psychological variables are very complex, and the study could consider more survey items from multiple aspects. Though we designed our questionnaires comprehensively, it is still possible that other survey items could measure certain latent factor. Third, this study could not draw causal relations between anti-epidemic campus signals and students’ preparedness because it is not a fully experimental design. It is still valuable for postsecondary administrators and policymakers to consider that this relationship indeed exists and that corresponding actions can, thus, be taken.

## 6. Conclusions

Preventing the transmission of COVID-19 is important for education [42]. This study sought to understand the effects of anti-epidemic campus signals on students’ COVID-19 preparedness. The results showed positive and significant relationships among these signals, awareness, and preparedness. In addition, we found that students’ COVID-19 knowledge can significantly predict their awareness and preparedness, and awareness has a positive association with preparedness. Notably, gender had an influence on students’ COVID-19 awareness and preparedness, while ethnicity did not. These findings provide valuable information for postsecondary administrators and policymakers to prepare for future pandemics.

## Figures and Tables

**Figure 1 ijerph-18-09276-f001:**
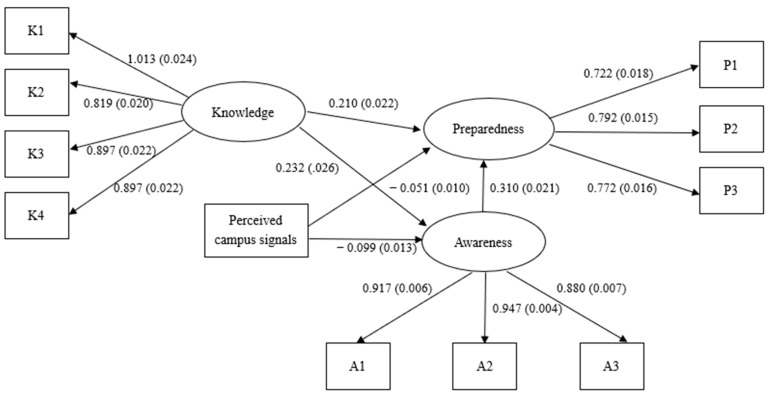
Final full structural regression model of students’ perceived anti-epidemic campus signals on their COVID-19 preparedness. Note: The factor loadings were reported by using standardized results, while the path coefficients were reported by using unstandardized results for an easier interpretation; standard errors are in parentheses.

**Table 1 ijerph-18-09276-t001:** Description of variables.

	Variable			Variable Description	Variable Type
	***Endogenous variable***				
		Awareness			
				Talk about COVID-19 with your classmates and/or roommates	Categorical Variable
				Talk about COVID-19 with your friends	Categorical Variable
				Talk about COVID-19 with your family	Categorical Variable
				*Items were measured by a 5-point Likert scales with 5 = Always, 1 = Never*	
		Preparedness			
				Wear surgical mask when going to class, attending school activities, etc.	Categorical Variable
				The frequency of using hand sanitizer	Categorical Variable
				Keep social distance when in public places	Categorical Variable
				*Items were measured by a 5-point Likert scales with 5 = Always, 1 = Never*	
	***Exogenous variable***				
		Perceived campus signals		Students’ perceived changes in anti-epidemic campus signals at the peak of the pandemic and post-COVID-19 era	Continuous Variable
		Knowledge			
				You know about the symptoms of COVID-19	Categorical Variable
				You know about how COVID-19 is spread	Categorical Variable
				You know about the anti-epidemic measures of COVID-19	Categorical Variable
				You know about the differences between COVID-19 and other pandemics	Categorical Variable
				*Items were measured by a 5-point Likert scales with 5 = extremely familiar, 1 = Not at all familiar*	
	***Control variable***				
		Male		Whether a student is male or not (1 = Yes, 0 = No)	Dichotomous Variable
		Han		Whether a student’s ethnicity is Han or not (1 = Yes, 0 = No)	Dichotomous Variable

**Table 2 ijerph-18-09276-t002:** Descriptive statistics of variables.

				Analytic Sample (*n* = 1464)			
	**Variables**			**Mean**	**SD**	**Min**	**Max**
	***Campus Signals***						
		Perceived campus signals		2.76	1.99	−2	7
	***Demographic Characteristics***						
		Male		0.53	0.50	0	1
		Female		0.47	0.50	0	1
		Han		0.94	0.24	0	1
		Others		0.06	0.24	0	1
	***Knowledge of COVID-19***						
		Know about COVID-19		2.93	0.92	1	5
		Know how COVID-19 is spread		3.07	0.93	1	5
		Know anti-epidemic measures		3.16	0.91	1	5
		Know differences		2.66	1.00	1	5
	***Awareness of COVID-19***						
		Talk about COVID-19 with classmates		2.37	0.80	1	5
		Talk about COVID-19 with friends		2.26	0.75	1	5
		Talk about COVID-19 with family		2.29	0.77	1	5
	***Preparedness of COVID-19***						
		Surgical mask		2.21	0.92	1	5
		Hand sanitizer		2.24	0.95	1	5
		Social distance		2.23	0.87	1	5

**Table 3 ijerph-18-09276-t003:** Correlations between variables for analysis of a structural regression model.

	Variables		1	2	3	4	5	6	7	8	9	10	11
	***Campus Signals***												
		1. Perceived campus signals	1.00										
	***Knowledge of COVID-19***												
		2. Know about COVID-19	−0.01	1.00									
		3. Know how COVID-19 is spread	0.01	0.84	1.00								
		4. Know anti-epidemic measures	0.00	0.79	0.87	1.00							
		5. Know differences	−0.07	0.73	0.73	0.69	1.00						
	***Awareness of COVID-19***												
		6. Talk about COVID-19 with classmates	−0.16	0.25	0.20	0.22	0.20	1.00					
		7. Talk about COVID-19 with friends	−0.22	0.22	0.16	0.19	0.21	0.87	1.00				
		8. Talk about COVID-19 with family	−0.20	0.24	0.18	0.21	0.20	0.81	0.83	1.00			
	***Preparedness of COVID-19***												
		9. Surgical mask	−0.18	0.22	0.19	0.18	0.31	0.34	0.36	0.34	1.00		
		10. Hand sanitizer	−0.14	0.26	0.24	0.28	0.35	0.32	0.37	0.37	0.58	1.00	
		11. Social distance	−0.19	0.27	0.23	0.25	0.36	0.30	0.38	0.36	0.56	0.61	1.00

NOTE: The correlations among Variables 2–11 are polychoric correlations due to the nature of ordinal variables.

**Table 4 ijerph-18-09276-t004:** A step-wise measurement model comparison procedure and an evaluation of the structural part of the SR model.

	Model	CFI	TLI	RMSEA	SRMR	*χ* ^2^	*df*	Δ*χ*^2^	Δ*df*
**Measurement models**									
1.	Initial model	0.990	0.988	0.081	0.051	444.86	42		
2.	Spreading mechanisms with anti-epidemic measures	0.991	0.989	0.078	0.050	403.14	41	41.72 *	1
3.	COVID-19 with Differences	0.992	0.989	0.076	0.048	374.37	40	70.49 *	2
**Structural part of the model**									
4.	Full structural model	0.998	0.997	0.038	0.022	119.67	38	254.7 *	2

NOTE: * *p* < 0.01. CFI = comparative fit index; TLI = non-normed fit index; RMSEA = root mean square error of approximation; SRMR = standardized root means square residual.

**Table 5 ijerph-18-09276-t005:** Full structural regression model of students’ perceived anti-epidemic campus signals on their COVID-19 preparedness with control variables.

				Model 1		Model 2	
	Paths			β	SE	β	SE
	***Paths to Preparedness***						
		1. Knowledge → Preparedness		0.210 ***	0.022	0.210 ***	0.022
		2. Perceived signals → Preparedness		−0.051 ***	0.010	− 0.048 ***	0.010
		3. Awareness → Preparedness		0.310 ***	0.021	0.312 ***	0.022
	***Paths to Awareness***						
		4. Knowledge → Awareness		0.232 ***	0.026	0.229 ***	0.026
		5. Perceived signals → Awareness		−0.099 ***	0.013	−0.093 ***	0.012
	***Controls***						
		6. Male → Preparedness				−0.113 **	0.038
		7. Male → Awareness				−0.163 **	0.050
		8. Han → Preparedness				0.048	0.070
		9. Han → Awareness				0.165	0.097
	***Model fit indices***						
		CFI		0.998		0.997	
		TLI		0.997		0.995	
		RMSEA		0.038		0.045	
		SRMR		0.022		0.021	

NOTE: ** *p* < 0.01, *** *p* < 0.001. CFI = comparative fit index; TLI = non-normed fit index; RMSEA = root mean square error of approximation; SRMR = standardized root means square residual.

## Data Availability

According to the data access policies, the data used to support the findings of this study are available from Zhejiang Academy of Higher Education, Hangzhou Dianzi University. Reasonable requests for CSEPPCE data can be made by email: zhaoteng@hdu.edu.cn.

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
