# Peer review of "Will Anti-Epidemic Campus Signals Affect College Students’ Preparedness in the Post-COVID-19 Era?"

_ijerph, 2021, doi:10.3390/ijerph18179276_

Round 1
Reviewer 1 Report
I consider the paper presented to me for review as very interesting and scientifically valuable. This study aims to exploring whether students’ perceived campus signals of anti-epidemic have effects on students’ awareness and preparedness of COVID-19. In this sense, I appraise this paper innovative and worth publishing. I congratulate the Authors of the research carried out. However, I have a few minor remarks that could help improve the quality of this well-written article.
I have no major reservations about the theoretical part of the article (“Introduction”). In my opinion, the literature review and theoretical justification of the research are sufficient.
I have some reservations about the "Method" section. Here are my suggestions. It is imperative that you add the section "Participants" to the "Method" section. Here, the participants of the research should be thoroughly described in terms of demographic variables.
In the "Analytic Plan" section, I think, it would be helpful to highlight a problem of common method bias which can appear in questionnaire-based research. This bias is caused by the common variance of applied measures, which does not come from investigated constructs but from the measurement method itself. Among others, common method bias may influence the parameters of the covariation between constructs.
Reviewer 2 Report
Anti-epidemic Campus Signals 081721
Overview: The authors surveyed full-time college students to determine whether perceived anti-epidemic campus signals affected the students’ awareness and preparedness for COVID. Per the authors, anti-epidemic campus signals were associated with awareness and preparedness for COVID. Female students had higher awareness and better preparedness.
Might “anti-epidemic campus signals” be better stated as “campus messaging, policies or practices designed to mitigate the epidemic.”
Introduction: It would be good in section 1.0 or 2.0 to more clearly define what is meant by anti-epidemic campus signals or policies. A table listing such items would be helpful to the reader. These items are mentioned in 3.2.2, but it is not clear how often surveyed students recognized specific campus signals.
The authors note that their data were “collected from the survey of College Students’ Epidemic Preparedness in Post-COVID-19 Era (CSEPPCE).” Provision of a reference to the survey items (e.g., online website link to the survey) and a brief description of the survey would be helpful for the reader. Further there should be clarity on when the survey was administered, how subjects were obtained, response rate to the survey, how were missing data addressed, etc.
Literature Review: When the authors speak of higher awareness of COVID, what is the actual measure of awareness used for the survey? Later in Table 1, they state that it was a Likert scale response to speaking to others (3 groups) about COVID.
Lines 117-120: It would be helpful for the authors to define the Han ethnic population and describe why they might be better prepared than female students - hypotheses (#5).
Lines 198-205: Again, it is not clear how often surveyed students recognized specific campus signals. Thus, it appears that the authors used the perceived change of campus signals as their independent marker for the campus signals. Thus, the authors could not account for whether some respondents were more perceptive of their environment than others and whether that perception could account for differences.
Lines 206-212: The authors did not actually measure the students’ knowledge of COVID, but rather asked how the students would rate their knowledge. Thus, the authors could not adjust for over- or under- confidence in one’s knowledge.
The model linking associations with perceptions of campus signals, COVID awareness, and COVID preparedness are interesting, but it would seem intrinsically that such perceptions should be closely linked due to colinear linkage.
The implications for these findings at other campuses require clarity.
Round 2
Reviewer 2 Report
I would still like to have more information about the original survey available to the reader.
Author Response
Thank you for your feedback! We attached our survey below. Our survey link is: https://www.wjx.cn/vj/ek625Le.aspx
The survey in WJX.CN is in Chinese because of the Chinese student participants. For the English journal review purpose, we attached both Chinese and English versions of CSEPPCE.
Feel free to let us know if you have further questions or concerns! Thank you!
